# Gait-Based Identification Using Deep Recurrent Neural Networks and Acceleration Patterns

**DOI:** 10.3390/s20236900

**Published:** 2020-12-03

**Authors:** Angel Peinado-Contreras, Mario Munoz-Organero

**Affiliations:** 1School of Engineering, Universidad Carlos III de Madrid, 28911 Leganés, Madrid, Spain; 100317846@alumnos.uc3m.es; 2Department of Telematic Engineering and UC3M-BS Institute of Financial Big Data, Universidad Carlos III de Madrid, 28911 Leganés, Madrid, Spain

**Keywords:** accelerometry, gait, walk, identification, recognition, recurrent neural network, LSTM, accuracy, smartphone

## Abstract

This manuscript presents an approach to the challenge of biometric identification based on the acceleration patterns generated by a user while walking. The proposed approach uses the data captured by a smartphone’s accelerometer and gyroscope sensors while the users perform the gait activity and optimizes the design of a recurrent neural network (RNN) to optimally learn the features that better characterize each individual. The database is composed of 15 users, and the acceleration data provided has a tri-axial format in the X-Y-Z axes. Data are pre-processed to estimate the vertical acceleration (in the direction of the gravity force). A deep recurrent neural network model consisting of LSTM cells divided into several layers and dense output layers is used for user recognition. The precision results obtained by the final architecture are above 97% in most executions. The proposed deep neural network-based architecture is tested in different scenarios to check its efficiency and robustness.

## 1. Introduction

The number of devices connected to the Internet (IoT) grows every year and consequently the data generated by them [1,2]. Among these devices, smartphones play an important role because of the continuous increase in their functionalities and user acceptance. Therefore, security in these devices must be one of the main points to be considered in their development. Some of the latest techniques used for preventing unauthorized access to mobile devices are based on biometrics. This involves the analysis and measurement of different physical characteristics or behaviors of an individual, with the objective of achieving recognition or identification.

Biometric identification methods have evolved over time and with the evolution of technology. The most well-known are fingerprinting, facial recognition, retinal scan, palm geometry, and voice recognition among others. However, other biometric variants are emerging that aim to be less intrusive. An example of this is the recognition of people based on the characteristics of their gait. This identification according to the way a person walks has a number of advantages and disadvantages. Its main advantages are that it allows an automatic, periodic, and non-intrusive identification (following the principles of “calm computing,” where the user is not disturbed by the technology), since he or she only has to carry the device (which will usually be a smartphone) or be recorded to make an identification by computer vision. As disadvantages we can mention that they are biometric methods that offer less accuracy than others like fingerprints. This can be solved by the continuous and periodic identification mentioned above. The performance of gait-based identification methods can also be boosted if the data samples used to perform the user detection are formed by a large enough interval of the user walking (for example 3 s as used in this paper), trying always to require the shortest time possible to perform a correct identification.

In this manuscript, user identification has been proposed through the use of recurrent neural networks (RNN), which use the acceleration patterns generated by users when walking, and which are captured by a smartphone sensor. These data are captured axially, and will be processed to calculate the vertical acceleration (or acceleration over the axis of gravity) as it simplifies the data and allows the identification to be done independently from the orientation in which the device was placed on each case. This data pre-processing and further development of neuronal architecture are done with Python 3 programming language.

The database that is being used is open access [3]. It is a complete database made up of 15 subjects (8 men and 7 women) who perform different activities while monitoring and capturing data with the devices and sensors (mostly smartphones) attached to their body. The subjects in the database have an average age of 32 ± 12.4 years, an average height of 1.73 ± 6.9 m and an average weight of 74.1 ± 13.8 kg. The smartphone used for the capture is the “Samsung Galaxy S4” as mentioned in the information about database. It is also important to mention that the activity has been carried out freely, without restrictions and in different environments (country, city, street...).

The main contributions of the paper are based on the following points. First, we carry out an identification of the users based on the accelerometry patterns, trying to achieve the highest possible accuracy, as well as a robust and efficient model in different scenarios. Second, we develop a deep neural network model, based on LSTM (“long-short-term memory”) layers adapted to the optimal processing of time series of data from wearable sensors. Third, we carry out experiments and tests to check the effectiveness and robustness of this model in different scenarios showing promising results.

## 2. State-of-the-Art

Different mechanisms and methods have been proposed and validated in previous research studies in order to recognize a particular person based on his or her gait characteristics. The mechanisms could be divided into three major families according to the technology used for gait monitoring and characterization: vision, floor sensor, and wearable sensor-based mechanisms [4]. Wearable sensors do not suffer from the hindering factors affecting computer vision methods such as occlusions, camera field of view/angle, or illumination deficiencies and are nowadays commonly embedded in widespread mobile devices providing scenarios for gait recognition systems without special equipment (either cameras or equipped floors) [4].

Marsico et al. [5] performed a detailed review of existing techniques for person identification based on gait parameters extracted from wearable sensors. The authors classify biometric traits for person identification as hard and soft. Hard ones generally allow a sufficiently accurate subject recognition, though suffering from trait-specific problems. Soft traits either identify entire groups of individuals or lack sufficient permanence. Although gait is considered a soft trait, the accuracy of the algorithms has significantly improved during the past decade. A second review on the use of wearable sensors for gait monitoring and person identification is presented in [6]. The authors present several taxonomies of human gait and its applications in clinical diagnosis, geriatric care, sports, biometrics, rehabilitation, and industrial areas. Available machine learning techniques are also presented with available datasets for gait analysis.

The challenge of person identification based on the time series captured from wearable sensor devices such as accelerometers and gyroscopes has been approached in several ways in previous research studies. Deb et al. [7] proposed a gait-based identification algorithm based on finding similarities on the wearable sensor data based on a new time-warped similarity. Comparing the time dynamics of the captured sensor signals with previous labeled data, the user with the highest similarity could be proposed as the identified user. Several metrics have been using for finding such similarities. The authors in [8] proposed a solution based on signal similarities based on correlation, frequency domain, and statistical distributions of the data acquired from a three-dimensional accelerometer, which was placed on the subject’s waist. The paper in [9] proposed a method for identification using absolute peak distance analysis, correlation, histogram, and first-order moments. Thang et al. [10] made use of time domain data to perform similarity analysis with “dynamic time warping” (DTW). The similarity calculations have been optimized for limited sensor device execution [11] and particularized for segments of users with particular needs such as elderly people [12].

A different approach for user identification based on wearable sensor data has applied machine-learning methods in order to find patterns and particular traits for each specific user. The authors in [10] extracted frequency components from accelerometer data and used support vector machines (SVM) in order to classify users. A different method based on the use of SVM with radial approximation was presented in [13]. The authors in [14] also proposed the use of SVM for person identification based on wearable sensor data. Other machine learning algorithms have also been proposed and validated. The paper in [15] used a Hidden Markov model for accelerometer-based gait recognition while the paper in [16] proposed the use of the k-NN (k Nearest Neighbor) algorithm in order to find the closet user in the space of computed features compared to previous traces recorded in the dataset. There have also been proposals to optimize the results by combining several algorithms and performing a majority vote at the end such as [17].

The idea behind machine learning models is to able to find particular patterns in the data with small variability for intra user data and big differences among users. In order to find intricate patterns embedded in the sensor data, deep learning algorithms have also been recently used with wearable sensor data for user identification. The authors in [18] explore the use of a convolution neural network (CNN) in order to extract the features from data that are fed into a one-class support vector machine (OSVM) for user verification. Giacomo et al. in [19] also make use of a deep CNN structure with a couple of fully connected layers and a softmax function in order to associate a probability of detection for each of the users in the dataset. The research paper in [20] presents an in-depth study of the building and training of a recurrent convolutional neural network with a real dataset based on gait reading performed through five body sensors. The time series from five body sensors are fed into a convolutional neural network with 2 1D convolutional layers and a recurrent neural network based on gated recurrent units (GRU). The output is passed into two pooling layers for feature sub-sampling and a final softmax function is used to estimate the likelihood for each data sample to capture the gait of a particular user. The authors in [21] also make use of a hybrid deep neural network based on both convolutional neural networks (CNN) and recurrent neural networks (RNN). Apart from the learning structure, the paper focuses on analyzing the data of free walking segments of the participants (not constrained to perform a particular activity in a predefined scenario). Recurrent neural networks (RNN) are able to model complex time-dependent patterns and have also been used in [22] for optimal feature extraction for gait characterization and user recognition. However, several mechanisms for manual tuning were used which limit the generalization of the results.

In the current paper, we explore and optimize RNN structures for gait recognition for users freely walking in the wild. Compared to the paper in [21] which explores deep learning techniques for unconstrained walking segments, we are able to improve the accuracy of result by almost 4%.

## 3. Proposed Methods

This section describes the pre-processing of the acceleration data, and explain in detail the method and architecture used to perform the identification based on the optimization of a deep recurrent neural network that will learn the particular patterns to characterize the gate particularities of each participant. 

### 3.1. Data Pre-Processing

The database that is being used is open access [3] and it is made up of 15 subjects (8 men and 7 women) who perform different activities while monitoring and capturing data with the devices and sensors (mostly smartphones) attached to their body. Among the activities, this section will focus on the analysis of the free walking data. The subjects in the database have an average age of 32 ± 12.4 years, an average height of 1.73 ± 6.9 m, and an average weight of 74.1 ± 13.8 kg. The smartphone used for the capture is the “Samsung Galaxy S4.” The activity has been carried out freely, without restrictions and in different environments (country, city, street...). The samples can be found for different activities and with the sensor placed in different areas of the body. We have selected the samples corresponding to the walking activity when the sensor is placed in the thigh/pocket area since it represents a common scenario simulating a common case in daily life (representing a natural position in which this device is usually worn, and which occurs in most cases). The walking activity has been performed by the different users for approximately 10 min, resulting in a total of 30,000–35,000 samples per user, so the sampling frequency will be around 50 Hz. This implies that the time between samples is 0.02 s, which is important to consider when packing the samples to represent a given time of activity.

#### 3.1.1. Calculating the Vertical Acceleration Component

The first step in pre-processing the data is the calculation of the vertical component of the acceleration (or acceleration about the axis of gravity). This procedure consists of making an estimate of the direction of the gravity force in a window with a duration of “2 m” samples. It means that the gravity vector will be estimated at each instant, using the previous “m” samples and the “m” following samples at the instant in which the estimate is to be calculated (see Figure 1). To do this, first the vector is calculated where each component corresponds to the mean value of the axial acceleration (X, Y, or Z) in a window of length “2 m” samples as shown in Figure 1. For this case, a window of size 2 m = 300 samples has been selected, since it has offered good estimation and results in user recognition. This window size is also selected since it is aligned with the optimal data frame length to be used as the input for the model as shown in Section 4.3.3. Windows smaller than 2 s show that there is not enough information in the sensor data to fully describe the particular walking patterns to characterize each user. A window of 3 s provides the optimal value which both captures the necessary information describing the walking activity with the minimum number of samples.

Once the vector of the mean acceleration value on the three axes has been obtained, $\left(\bar{x_{m}},\bar{y_{m}},\;\bar{z_{m}}\right)$, the module of the vector is calculated, obtaining $\left|\bar{X_{m}},\bar{Y_{m}},\;\bar{Z_{m}}\right|$:(1)$$\left|\bar{X_{m}},\bar{Y_{m}},\;\bar{Z_{m}}\right|\;=\;\sqrt{{\bar{X_{m}}}^{\;2}+{\bar{Y_{m}}}^{\;2}+{\bar{Z_{m}}}^{\;2}},$$

The unit vector of gravity $\hat{g}$ is then estimated as:(2)$$\hat{g}=\frac{\bar{X_{m}},\bar{Y_{m}},\;\bar{Z_{m}}}{\left|\bar{X_{m}},\bar{Y_{m}},\;\bar{Z_{m}}\right|}=\left(g_{x},g_{y},g_{z}\right),$$

To obtain the acceleration on the axis of the estimated gravity force at each instant (*acc_g_*), a scalar product is made between the value of acceleration captured at that instant and the estimated unit vector of gravity
(3)$$acc_{g}=acc_{\left(X,Y,Z\right)}\;\cdot \;\hat{g}\;=\;\left(acc_{X},\;acc_{Y},\;acc_{Z}\right)\cdot \;\left(g_{X},\;g_{Y},\;g_{Z}\right)=\;acc_{X}\;\cdot g_{X}\;+acc_{Y}\;\cdot g_{Y}\;+acc_{Z}\;\cdot g_{Z},$$

This value of the estimated acceleration on the axis of gravity is less dependent on the orientation of the device and has a more representative value on the way the user moves while walking. Some graphs of this value can be shown for different users in Figure 2.

The initial and final moments of these vertical acceleration values ($acc_{g})$ have been eliminated, because they correspond to the moments when the user started/stopped walking and the acceleration is so low that it practically represents noise.

#### 3.1.2. Sample Packaging

Sample packaging means that the vertical acceleration samples ($acc_{g})$, are grouped together to represent a specific time of user activity. If a single acceleration sample were used for identification, it would be virtually impossible for the neural network to perform user identification. For the general development of this article, a package of 150 samples has been used, as shown in Figure 3, so that each sample package represents an approximate time of 3 s of the user walking (150 samples • 0.02 s/sample = 3 s). We have tried to keep it as short as possible to speed up the identification and to avoid high capture times for user recognition, always keeping a high precision as the main objective.

#### 3.1.3. Data Labeling for Training and Testing the Algorithm

Each user’s samples have been labeled with a numerical identifier. This way, each vertical acceleration sample can be associated with a user and know who it belongs to. After packing the samples in windows, each sample pack will be labeled, so that it can be assigned to its respective user. This will be necessary to run the training and validation of the neuronal model.

A categorization of the label has also been carried out. This procedure consists of making a one hot encoding (Figure 4), so that the label is represented by a numerical vector with 0 and 1 values.

The last layer of the neural network will be made up of as many neurons as there are subjects to be identified in the database. The activation function in the last layer will be softmax so that, for each sample, the output is a vector of the same shape and length as this categorization. Therefore, it is possible to obtain the estimate that the neuronal network makes on the samples and assign it to the corresponding user after making the inverse transformation of the categorization.

#### 3.1.4. Sample Train-Test Division

Finally, in order to perform a proper model training and evaluation, it will be necessary to divide the data set into training and test. To do this, a random partition of 70% of the data is made for training and 30% for testing. The training data are used to model the neural network and make it learn how to identify each user when introducing the corresponding samples. The test data are used so that once the complete model has been trained it can be validated and its performance checked with samples that the system has not used during training, and which are therefore new to the neural network.

### 3.2. Proposed Architecture

#### 3.2.1. Recurrent Neural Networks and LSTM Cells

A deep recurrent neural network has been designed to carry out the identification based on LSTM cells. Previous architectures described in Section 2 made a more extensive use of a different architecture based on deep convolutional neural networks. Recurrent neural networks are designed to be able to learn patterns from time series. The research in this manuscript has been based on the study of such deep architectures in order to characterize the gait of different people in order to perform user identification. 

In a summarized way it can be said that LSTM (“long-short-term memory”) neural networks use memory units in order to remember values from the short and long past data reacting to time patterns for each user. Each LSTM cell is composed of a status cell and three doors: entrance door, exit door, and forget door. 

The status cell can be interpreted as a continuous flow of information over several instants of time. At each instant of time you have to decide which information is retained, which is modified, and which is allowed to pass.Entrance door: Controls when new information can enter the status cell. It also determines how much of that new information will be allowed into the status cell.Exit door: Controls when the information from the memory is used for the result and how much information from the status cell will be offered at the output.Forget door: Controls when information is forgotten. This way, you can create space for new important data and remove less relevant ones. The function of this door is to determine whether the information will be stored in the status cell, or whether it will be discarded.

The details for LSTM networks can be found in [23,24].

#### 3.2.2. Proposed Architecture for User Identification Based on Gait Data

The proposed architecture uses a stack of RNN layers connected to several dense layers and an output softmax layer in order to associate a probability to each window of input data to belong to each user in the system. The parameters and different options of the network have been adjusted between executions in order to optimize the accuracy of results.

The details for each of the layers are:The first two layers are RNN based on LSTM cells, formed by as many LSTM cells as there are moments of time (samples) in the packaged input data. Since the samples have been packed in windows of 150 samples (3 s), there will be 150 LSTM cells processing each time window of input samples. The first one has shown optimal results for 50 to 100 memory units (offering more or less similar results), and the second layer LSTM has shown better results for 100 memory units (as captured in Section 4). Memory units represent the internal state of the LSTM cell and are able to learn and forget time-dependent patterns based on the previous state and current inputs. The optimal number of memory units will therefore depend on the complexity of the time dependencies in the patterns to be detected.Then two hidden layers will be stacked, fully connected, with 100 and 90 neurons respectively. The activation function of these layers will be ReLU, as it offers a fast convergence that is especially useful in deep neural networks. This activation function usually works very well in the intermediate hidden layers.Lastly, there is a layer formed by as many neurons as the number of total users you want to differentiate. Since in this case the database has 15 users, there will be 15 neurons. The activation function for these neurons is softmax. As this is a multi-class classification problem, with this activation function it will be possible to obtain a categorical probability distribution over the different possible K-subjects.

The network design and development has been made in Python 3 programming language, because of its high compatibility with libraries such as Keras [25], which allows the implementation of neural networks. 

The following parameters of interest have been used for the training of the network:Loss function (the function used to estimate how far the current parameters are from the optimal ones when training the model and therefore the function to be minimized in such training): Categorical cross entropy. Used for multi-class classification problems.Optimizer (the numerical algorithm used to find the optimal value for the parameters in the model that minimize the loss function): Adam. It has the advantages of other optimizers such as RMSprop and SDG (Stochastic Gradient Descent) with momentum. It offers fast convergence and its operation is based on the use of first and second moment gradient estimates, adapting its learning rate to each weight of the neural network.Metric (the mechanism used to measure the achieved performance): Categorical accuracy. Accuracy of sample identification in multi-class classification problems.

## 4. Results, Testing, and Experiments

### 4.1. Results and Precision Test

We have tried different configurations and parameters in order to optimize the number of memory cells and the number of hidden layers. Some of the results and tests done with the variations of the proposed architecture are shown in Table 1, where the accuracy in the same test set and with different configurations can be seen:

All configurations offer very positive results in accuracy (above 95%). The use of multiple hidden intermediate layers has made the model more robust in accuracy and has a better convergence during training. The best results are for the ID1 and ID5 architecture, which corresponds to the configuration shown in Figure 5. To better measure the performance of these ID1 and ID5 configurations, we took different test groups and calculated the average accuracy in different executions, obtaining the results in Table 2. A different seed was used in the random algorithm to split the data into 70% training and 30% validation. The seed values used are captured as identifiers in the first column in Table 2. The results in Table 2 show the mean value and the variance for each configuration and group. The variance for the different groups for configuration ID5 tends to be smaller while the mean value shows an improvement around 1% in accuracy.

Generally speaking, the ID5 configuration is more accurate than the ID1 configuration, although it requires more training time because of the increased memory in the first LSTM layer.

Tests were also performed in which acceleration patterns are captured by a smartphone placed on the chest instead of inside the pocket. This type of data usually contains higher noise values because of involuntary movements, as well as being less representative of the user’s gait, so accuracy is expected to be lower than data from the user walking with the smartphone in the pocket.

Even so, positive precision results are obtained with an average that is above 95% in different test groups as shown in Table 3.

### 4.2. Training Graphics and Confusion Matrix

The training graphics and the confusion matrix are captured in this sub-section based on the architectures described in the previous one.

For the training graphs what is done is to look at the average accuracy of the model in each period. In this case, you can see the evolution and convergence of the models during the training phase. In addition, in this case 5% of the training data has been selected and a new set of validation data has been created, which will not be used for training. This is intended to see the real evolution of accuracy during training (in each epoch) on a set of data that is not used for training. This does not correspond to the actual estimated accuracy value. To see the training graphs, two runs have been made, one for each model, on the same training, test, and validation sets. These representations are shown in Figure 6 and Figure 7 for a total of 75 training epochs. First, for the ID 1 architecture the results are shown in Figure 6.

The same for the ID5 architecture are captured in Figure 7.

As shown, both the accuracy and the loss in each epoch are represented for the training set and the validation set. Accuracy is understood to be whether the model’s predictions of the samples match the actual label to which that sample belongs. Loss is the scalar value that tends to be minimized, since a lower value implies that the values predicted by the model are correct and correspond to the real labels of the samples. In this case, the categorical cross entropy loss function (“Categorical cross entropy”) has been used to calculate the value of loss. The two graphs are related because as accuracy increases, the loss will decrease.

Looking at both graphs, it can be seen how the convergence is faster in the case of using LSTM cells with 100 and 100 memory units (for layers one and two). In addition, in both architectures, stability is quickly reached around a high value for accuracy, however, more training epochs have been used to make the model robust and able to reach accuracy values like those obtained in the previous tables.

The confusion matrix is a tool to measure the performance of the classifier. It is a table where the values estimated by the neural network and the real values are shown. With this representation one can see which individuals are being confused by the neural network, which are the best identified, how many samples of each subject have been classified correctly or wrongly, etc. Figure 8 is an example of a run on a test set with the ID 5 architecture. In this particular case the accuracy obtained is 97.4%.

### 4.3. Generalization of Results for the Selected (Best) Neuronal Architecture

This section shows different experiments and validation tests carried out on the final architecture showing its robustness and efficiency in different areas.

#### 4.3.1. Increasing the Database with Additional Data

We propose to add to the database a series of external data generated in a personal way with the help of a smartphone. This is to assess whether the model is able to deal with other external data recorded with different hardware in a different setting and that its domain is not only restricted to the data in the database [3]. The following elements are used for this purpose:Smartphone with accelerometer and gyroscope sensors. In this case, the Xiaomi Redmi Note 4 smartphone is used, which includes these sensors, in particular the LSM6DS3 sensor, which performs both tasks and is available on many other devices. It is a sensor with a high sampling rate (around 200 Hz), so it can be interpolated and adjusted so that samples are taken at 50 Hz as available in the original database. This sensor must be able to capture the axial information in the X-Y-Z axes, so that the pre-processing mentioned before can be done in a similar way to the rest of the data already existing in the database.Data capture application. Must allow data export to CSV format and logging from multiple sensors simultaneously. In this case the application called “Sensor Data Collector” has been used, created by the database developers [3] and that is available in the corresponding GitHub repository [26].

Once these elements are available, it will be necessary to record the walking activity for 10 min by placing the mobile device in the same position as in the original database [3]. For this experiment we will also try to recreate as reliably as possible the activities performed in the original database [3]. Since the main results obtained are for the case of walking with a sensor in the pocket/thigh, the same dynamics will be followed. Therefore:A walking activity was recorded in two subjects (male and female) for 10 min continuously and without interruption.Because of the first wave of the COVID-19 pandemic that occurred in March–June 2020, it was impossible to record such activity in an open environment. For this reason, this activity was carried out at home with the help of a treadmill set at 4 km per hour.The data provided by the accelerometer and gyroscope were recorded simultaneously and synchronously.The CSV files generated for the two users were added with the rest of the database (only with the data of the thigh/pocket sensor walking activity). These will count as two new subjects with identifiers 15 and 16, which are added to the fifteen that previously had identifiers from 0 to 14. A total of 7139 samples are available (after pre-processing the vertical acceleration calculation, packaged in 150 and half-segment overlap), corresponding to 4990 for training and 2149 for testing.In the last layer of the neural network it is necessary to expand the number of neurons by two in order to be able to identify the 17 subjects available in the database after adding the new data. The activation function remains “softmax” since the purpose of identification is the same.

To check if the neural network is capable of identifying new users, training and evaluation are carried out on the ID 5 architecture that usually offers the best results. The results obtained after several executions show an accuracy above 95% in all cases. Figure 9 is the confusion matrix corresponding to an execution where the 96.78% accuracy has been obtained.

As we can see, the test samples corresponding to the two new subjects (15 and 16) have been correctly classified in a large percentage. In total 4 samples of subject 15 were confused with subject 16, and only 1 sample of subject 16 was confused with subject 15. Despite the fact that both cases were recorded under the same conditions, on the same surface and with the speed set at 4 km per hour, it was possible to make a correct identification of both subjects. The rest of the users are not affected by the introduction of new subjects and have positive identification in most of their samples. For the following sections, these two new subjects are included in the database as they provide a better view of the system’s performance over a wider range of data. This means that the model is robust to data generated by other mobile devices and in other environments. In addition, it is tested for correct operation outside the database, so it could work in different environments and with different mobile devices as long as they meet the requirements mentioned at the beginning of this section.

As a future work, the model will be tested with a wider range of users using different mobile devices in order to have further insights about the scaling of the model for a large number of users. The results presented in this section provide positive outputs for the identification of new participants when using different hardware in different settings since the confusion matrix shows that the walking segments for the 2 new users are never classified as belonging to the previous users in the dataset, and only one segment in the dataset in [3] is classified as being from one of the new users. However, an experiment with a wider range of users will provide further results in order to assess the scaling of the model.

#### 4.3.2. Performing User Identification with Gyroscope Extra Data

In this case, the objective is to check whether the system would improve its performance by using extra information besides the one provided by the accelerometer. This will be done using the data captured by the gyroscope during the activity. These data are axial in the X-Y-Z axes as they were in the accelerometer sensor, so they must be pre-processed in the same way. To do this, the vertical rotation, or also called rotation on the axis of gravity, is calculated.

The process is similar to that for acceleration data. First the direction of the gravity force is estimated by averaging the acceleration data as previously described. Then, the vertical rotation speed (the rotation over the direction of the estimated unit gravity force) is calculated using the dot product as:(4)$$w_{g}=w_{\left(X,Y,Z\right)}\;\cdot \;\overset{︷}{g}\;=\;\left(w_{X},w_{Y},\;w_{Z}\right)\;\cdot \;\left(g_{X},\;g_{Y},\;g_{Z}\right)$$

This value is used for user identification. Since it was recorded simultaneously to the accelerometer, it is possible to pack it in equal segments of 150 samples and both describe 3 s of the user walking in the same moments (as captured in Figure 10).

To check the impact on performance when adding this vertically oriented variable, a new test set is selected and three different runs are performed on the same test set. One with only the acceleration variable (as it has been done so far), one with the orientation variable, and one with both variables. Table 4 shows the average accuracy obtained in the different tests.

As expected, the accuracy increases when the information from both sensors is used. By having more information from each sample, the network is able to increase the accuracy and identify those samples where a single sensor fails. However, the difference is not so noticeable, only 2.1% between the case of the accelerometer and the case of both sensors. This test shown in the table is particular for one test group, but a similar behavior has been observed in the rest of the simulations with other test groups. In some isolated cases, greater accuracy has been obtained by using only the data provided by the accelerometer. This could be used in a complementary way to increase the accuracy, since generally the accelerometer and gyroscope data are provided by the same sensor, so their capture can be done simultaneously without problems. However, by using more variables, training takes longer, so this will also be an important aspect to consider.

#### 4.3.3. Changing the Window Size for Data Analysis

This section aims to see the effect of changing the packaging of data into sample segments. As previously mentioned, for the results shown throughout the document, a packaging of 150 acceleration samples was being used. The use of this window has been decided on the basis of different executions and observation of the results. We have tried to keep it as small as possible to speed up the identification and to avoid high capture times for user identification, always keeping as a main objective to achieve a high precision.

To see the effect of changing this size, runs with different data windows are performed and the accuracy obtained is observed. The tests are performed using data windows smaller and larger than 150. The neural network is the one already mentioned in several sections, ID 5 architecture, and in order to be able to compare fairly, the same set of test data is used for all executions. The training is done following the previous guidelines, so 100 training periods and a batch size of 32 are used. The results are shown in Table 5 (the training time has been included in the table to give an idea about the complexity of the model).

As expected, a reduction in the number of samples in each segment makes user identification more difficult. By using a smaller data package, the available information on the data window is not enough to fully characterize the walking activity, presenting the neural network with less data on each sample and therefore increasing the difficulty to identify a particular user as compared to the rest of the users. As can be seen in the table, the architecture is capable of identifying users with 80% accuracy when the samples are made up of segments of only 0.4 s, which can be considered as a really short time. The same happens for the case of 0.24 s per segment, where the accuracy is close to 70%. Convergence time during training slightly increases for segment sizes of 12 and 20 samples, as compared to segments of 50 or 100 samples, since more epochs are required to converge.

You can also see in Table 5 how the increase in segment size over 150 samples does not produce a noticeable improvement in performance. Therefore, it does not make sense to use data segments larger than the one used for this case (150), since the objective is to perform quick recognition by minimizing the walking time needed for identification.

The use of different sizes for the data segments also implies a modification of the architecture of the LSTM cells available in each LSTM layer, since the time dimension of the input data are modified according to this segment size. For the realization of this test, the architecture is modified in each execution so that each LSTM layer has as many as LSTM cells as the temporal dimension of the input data (Column of Table 5: “Packet data size”).

#### 4.3.4. Identification of Users in Other Activities

It has already been shown that user recognition works correctly for the case of walking activity. This is the most typical case and the one that occurs most naturally in any user. However, to test how effective the model is in other environments, we have carried out other experiments in which the activity from which the accelerometry patterns are captured is different. This had been done by using the data from the original database [3] (composed of the 15 users) and selecting the data corresponding to the activities: climbing stairs, running, and jumping. The files corresponding to the capture of the activity with a device placed in the thigh/pocket have been chosen for consistency with the rest of the experiments carried out in this work.

After processing the axial information of the files in the same way, the estimated acceleration on the axis of gravity (vertical acceleration) is calculated. To check if the proposed models are good in these activities, a precision test has been performed as in the different previous sections. A set of data for training and a set for testing were chosen, maintaining the proportionality of 70–30% of the data available for each activity. Training consisted of 75 periods with a batch size of 32 samples. The ID 5 architecture has been chosen for the different runs and the results are shown in Table 6.

For the different scenarios shown in the table, an accuracy above 85% is obtained, which allows the correct identification of the users in the different activities. As in the previous experiments, segments of 150 samples (3 s of activity) have been used for the input. Relatively good results are obtained in terms of accuracy without the need to modify an architecture that was initially designed for the walking activity. This shows the ability of recurrent neural networks and, specifically, of architectures designed to be adapted to different physical activities of the user, to perform identification in more scenarios and without depending on the user having to execute a particular single activity.

## 5. Conclusions

The results presented in this manuscript show that deep recurrent networks are able to automatically characterize the gait properties and particularities for users when walking in the wild (without the need of a predefined setting) with accuracies higher than 0.97. The results outperform previous studies for gait characterization in unconstrained conditions [21] in which the accuracy values are below 0.94 using a hybrid deep neural architecture and previous RNN-based architectures for gait inspection [22] in which the authors achieve accuracy values below 0.93. Although the accuracy of the proposed method in this manuscript only slightly outperforms other previous studies in controlled settings, it allows the walking activity to be freely executed by each user. The authors in [19] reported a 95% accuracy when the data are collected from six predefined walking segments. The authors in [20] obtained similar accuracy values from the ones presented in this manuscript but also using the same dataset in the same controlled environment for data capturing. 

The manuscript shows the optimization process for some of the major parameters in the architecture which provides optimal results for two RNN layers based on LSTM cells with 100 memory cells. 

The proposed architecture has also shown good generalization properties when including the information of new users recorded in sort of similar conditions. 

The results also show a slight improvement of results when using both the acceleration and the gyroscope sensors together, although the complexity of the algorithm and the required time for training also increase when adding the gyroscope data. 

The proposed network has also been trained for other activities such as running and jumping but the results for user identification are worse than for the case of walking (showing that the way we walk better characterize each person than the way we run, jump or climb up stairs).

## Figures and Tables

**Figure 1 sensors-20-06900-f001:**
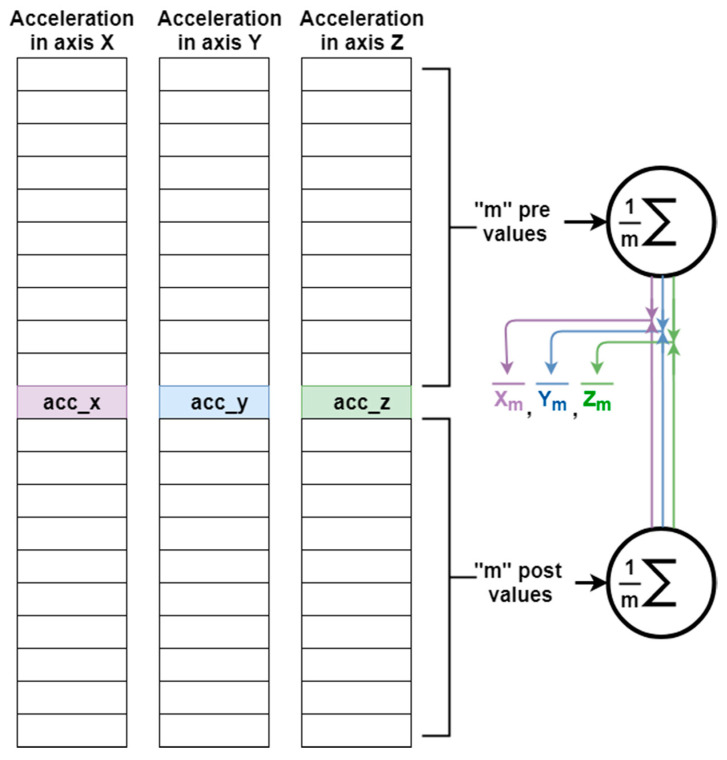
Calculation of the mean acceleration for the X, Y, and Z axes in a “2 m” length window.

**Figure 2 sensors-20-06900-f002:**
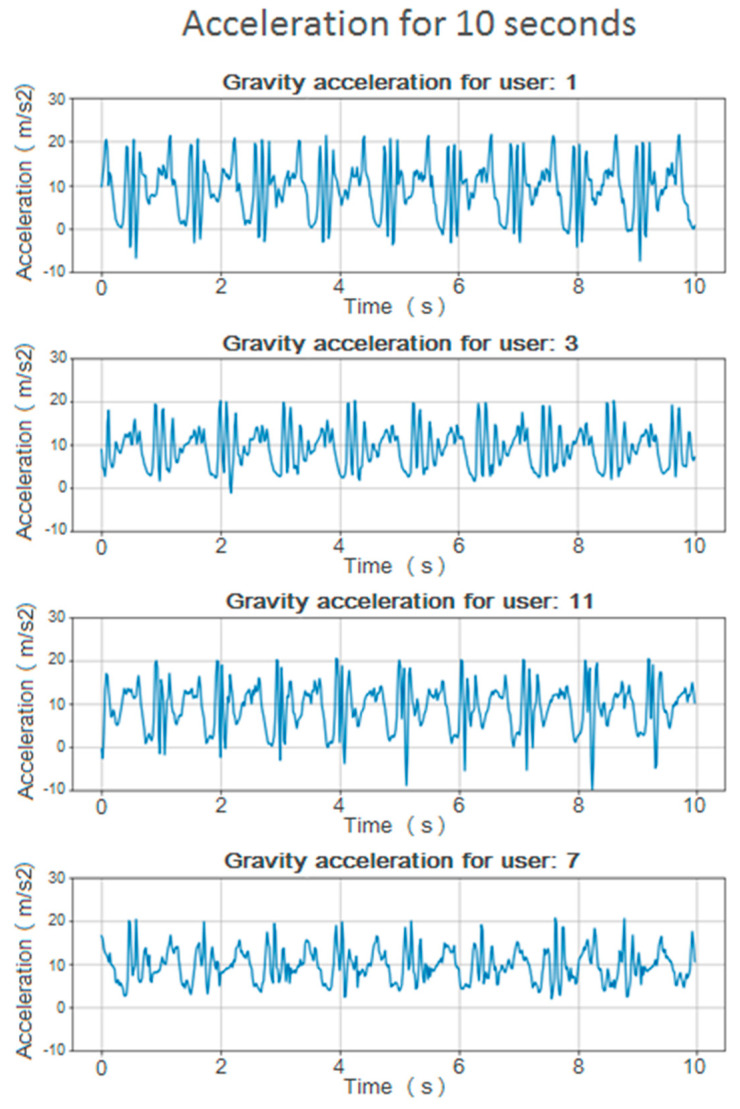
Vertical acceleration graphics for multiple users during 10 s.

**Figure 3 sensors-20-06900-f003:**
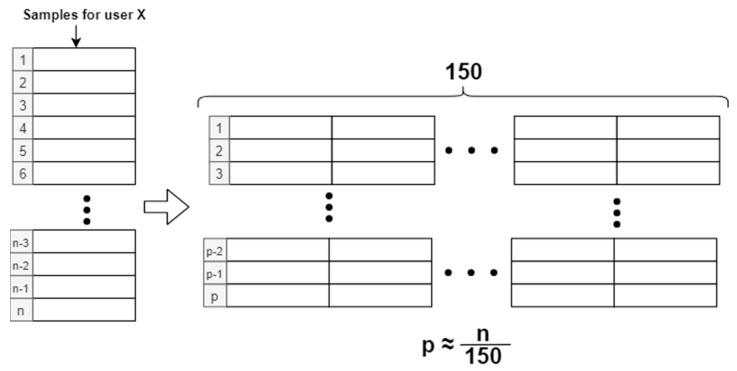
Window packaging of 150 samples.

**Figure 4 sensors-20-06900-f004:**
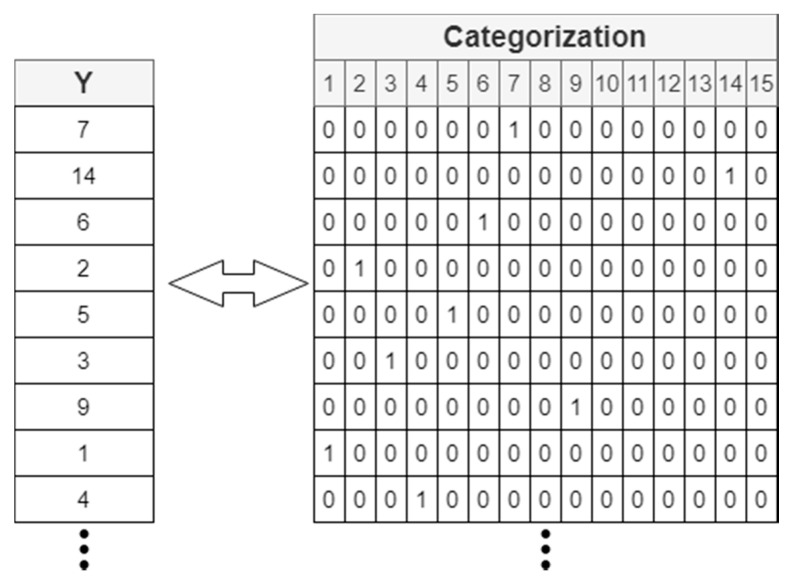
Label categorization with one hot encoding.

**Figure 5 sensors-20-06900-f005:**
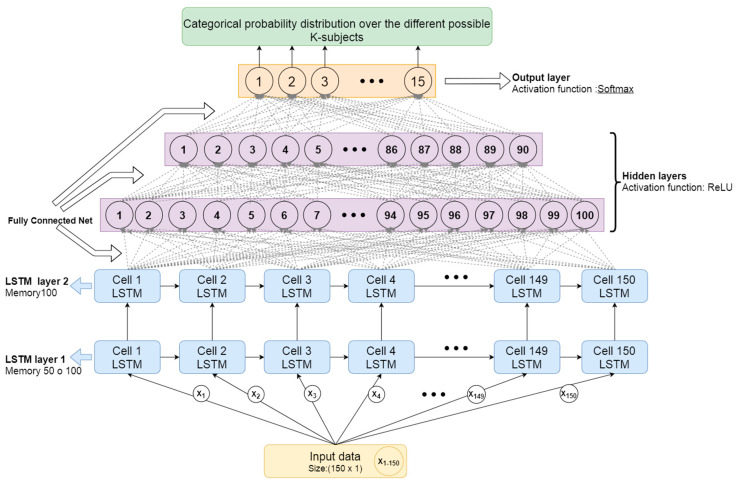
Neural network architecture designed for the identification.

**Figure 6 sensors-20-06900-f006:**
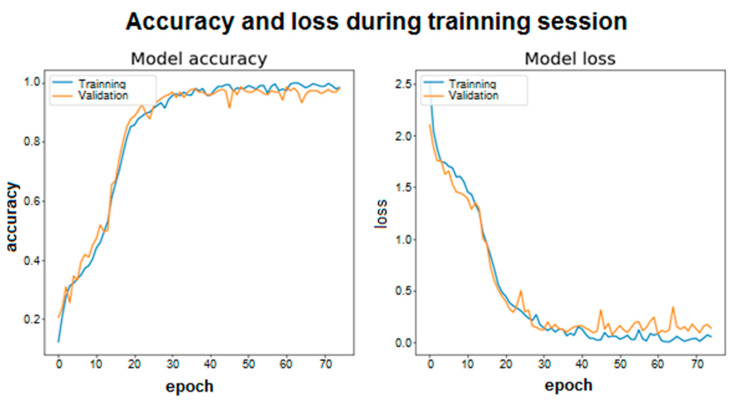
Accuracy and loss during training session ID1.

**Figure 7 sensors-20-06900-f007:**
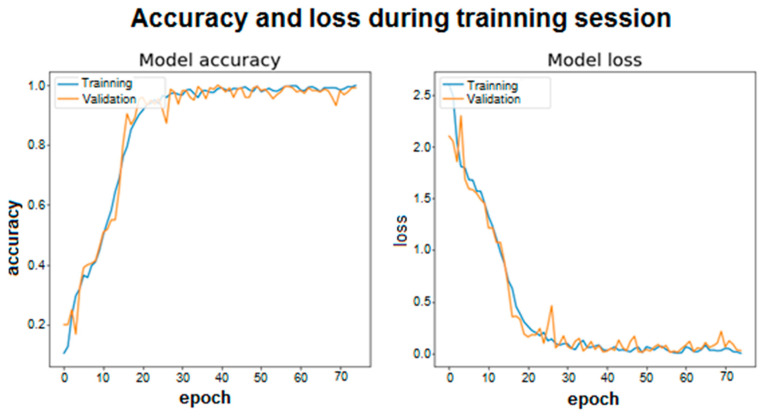
Accuracy and loss during training session ID5.

**Figure 8 sensors-20-06900-f008:**
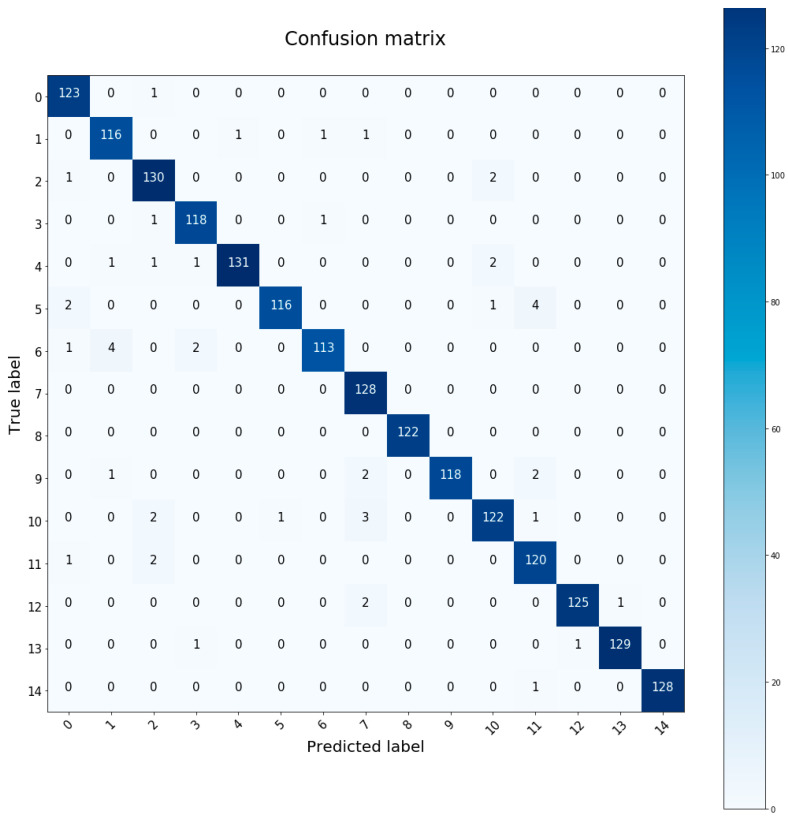
Confusion matrix with ID5 architecture. Execution with 97.4% test accuracy.

**Figure 9 sensors-20-06900-f009:**
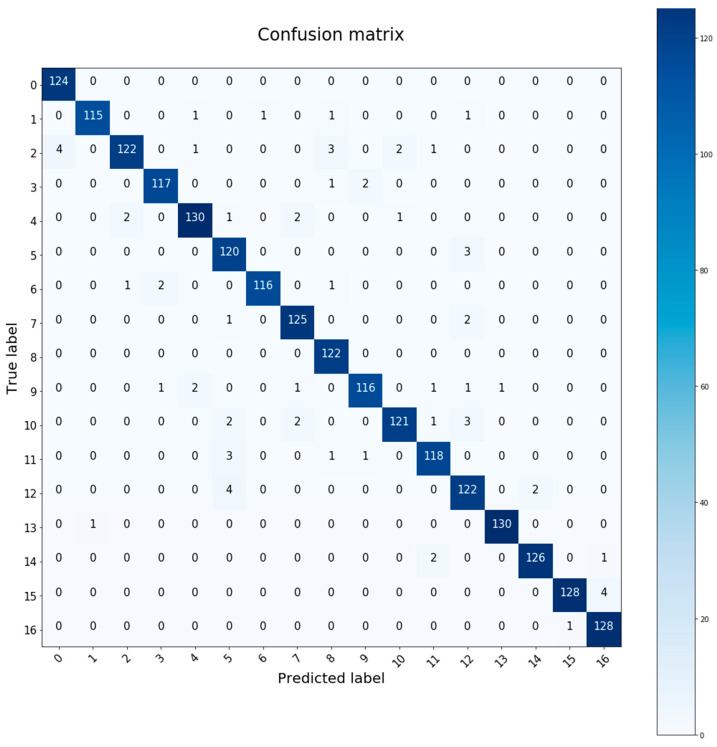
Confusion matrix for a test set. Extended case with two new subjects 15 and 16.

**Figure 10 sensors-20-06900-f010:**
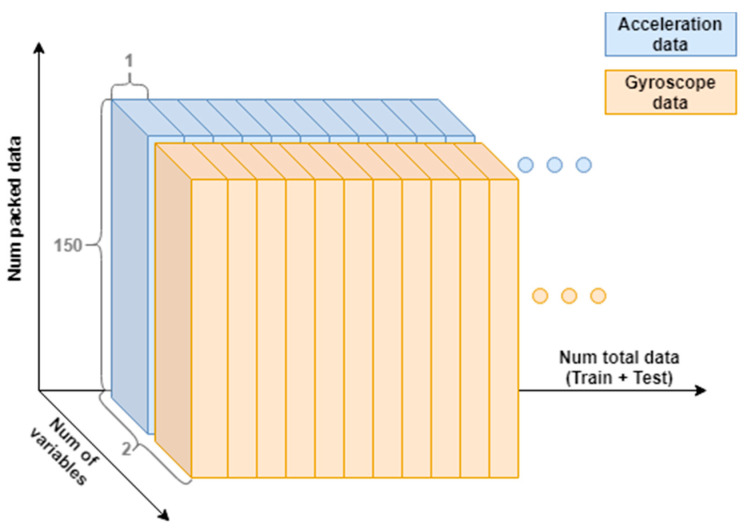
Packaging of the data formed by the two sensors (accelerometer and gyroscope).

**Table 1 sensors-20-06900-t001:** Precision results collected for different neural network architectures.

ID	Memory Units in LSTM Layers	Number of Hidden Layers	Number of Neurons in Hidden Layers	Number of Trainable Parameters	Accuracy Test Execution 1	Accuracy Test Execution 2	Accuracy Test Execution 3	Training Time
1	50 and 100	2	100 and 90	91.355	97.86%	98.14%	96.76%	~29 min
2	50 and 100	2	100 and 60	87.875	96.13%	96.87%	97.45%	~30 min
3	50 and 100	3	100, 90, and 60	96.365	96.34%	95.33%	95.18%	~31 min
4	50 and 100	3	100, 60, and 30	89.255	96.29%	95.49%	96.12%	~32 min
5	100 and 100	2	100 and 90	141.755	97.88%	97.61%	97.24%	~37 min

**Table 2 sensors-20-06900-t002:** Comparison of ID1 and ID5 architecture in different test sets.

Seed Group Test	Accuracy for Configuration ID 1 LSTM (50–100) and 90–100	Accuracy for Configuration ID 5 LSTM (100–100) and 90–100
10	Mean: 97.82% Variance: 0.861	Mean: 98.35% Variance: 0.299
20	Mean: 96.23% Variance: 0.98	Mean: 97.61% Variance: 0.667
40	Mean: 97.08% Variance: 0.612	Mean: 98.15% Variance: 0.425
50	Mean: 96.76% Variance: 0.774	Mean: 96.97% Variance: 0.746

**Table 3 sensors-20-06900-t003:** Accuracy in test data with chest sensor.

**Configuration ID 5**	**Accuracy in Test Data with Chest Sensor**	**Mean Accuracy**
	95.39%	95.793%
	96.12%	
	95.88%	

**Table 4 sensors-20-06900-t004:** Accuracy in test data when using different sensors.

Used Variables	Accuracy in Test Data
Only acceleration data	95.85%
Only gyroscope data	93.55%
Both acc and gyro’s data	97.95%

**Table 5 sensors-20-06900-t005:** Accuracy testing for different data packet sizes.

Packet Data Size	Equivalent Time of Each Packet	Number of Train Data	Number of Test Data	Accuracy in Test Data	Training Time
300	6 s	2481	1075	98.14%	48.3 min
250	5 s	2983	1291	97.90%	34.2 min
200	4 s	3735	1613	96.40%	32.7 min
150	3 s	4990	2149	98.09%	35.56 min
100	2 s	7498	3223	95.03%	32.4 min
50	1 s	15,022	6448	93.24%	32.6 min
20	0.4 s	18,797	8063	80.14%	33.1 min
12	0.24 s	31,322	13,440	68.58%	33.3 min

**Table 6 sensors-20-06900-t006:** Accuracy in test data for different activities.

Activities	Number of Data Train/Test	Training Time	Accuracy in Test Data
Climb stairs	3622/1560	~26 min	90.06%
Run	4883/2101	~35 min	87.91%
Jump	538/239	~3.5 min	90.79%

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
