# Peer review of "Gait-Based Identification Using Deep Recurrent Neural Networks and Acceleration Patterns"

_sensors, 2020, doi:10.3390/s20236900_

Round 1

Reviewer 1 Report

I have attached an annoyed copy of the manuscript. In summary, this is an interesting paper investigating the question of biometric identification by recognising gait with an RNN. The topic is not novel, but the CNN architecture is. The paper is clearly written, with some minor errors in the English, as indicated. Some details have been omitted, such as the number of recordings used for training and testing.

Queries that should be addressed in a resubmission are:

Does this method scale up for a general purpose person identification system? Showing it is effective with 17 subjects is insufficient for this purpose.

Section 3.1.1 states that windows of 300 samples are used, but there is no justification for this choice. The section also talks about "making an estimate of  the value of gravity", which confused me, hence my comment on figure 2 querying why it isn't 9.8m^-2? The author should clarify that they are measuring the vertical component of instantaneous acceleration.

Figure 2 could be improved by labelling the axis with time (in seconds). The timestamp is less informative. 

Page 10 - what are the "different test groups"

Table 2: you could compute the mean and st dev of the measurements and assess how significant is the difference between the two columns?

Page 16: Is training time really important, after all, you only train once.

Some other comments are indicated on the document.

Author Response

Answer to the reviewers’ comments for the paper: Gait Based Identification Using Deep Recurrent Neural Networks and Acceleration Patterns

We would like to thank the reviewers for their positive feedback which has allowed us to improve the quality of the manuscript. All the proposed changes have been fully revised and thoroughly implemented.

Reviewer 1

Comment:

Does this method scale up for a general purpose person identification system? Showing it is effective with 17 subjects is insufficient for this purpose.

Answer:

Section 4.3.1 has been expanded to deal with this comment. The reviewer is right the adding more users in different settings with different sensor devices will provide further insights for the scaling of the model. The results presented in the paper for the 2 new added users provide some interesting intuition and we have combined all the above in a new paragraph at the end of section 4.3.1 to better describe what has been done and what will be done in future work:

As a future work, the model will be tested with a wider range of users using different mobile devices in order to have further insights about the scaling of the model for a large number of users. The results presented in this section provide positive outputs for the identification of new participants when using different hardware in different settings since the confusion matrix shows that the walking segments for the 2 new users are never classified as belonging to the previous users in the dataset, and only one segment in the dataset in [3] is classified as being from one of the new users. However, an experiment with a wider range of users will provide further results in order to assess the scaling of the model.

Comment:

Section 3.1.1 states that windows of 300 samples are used, but there is no justification for this choice.

Answer:

The following text has been added in order to further justify the selected values:

This window size is also selected since it is aligned with the optimal data frame length to be used as the input for the model as shown in section 4.3.3. Windows smaller than 2 seconds show that there is not enough information in the sensor data to fully describe the particular walking patterns to characterize each user. A window of 3 seconds provides the optimal value which both captures the necessary information describing the walking activity with the minimun number of samples.

The size of the window and its implications in the results is captured in the table:

Packet data size

Equivalent time of each packet

Number of train data

Number of test data

Accuracy in test data

Training time

300

6 seconds

2481

1075

98.14 %

48.3 min

250

5 seconds

2983

1291

97.90 %

34.2 min

200

4 seconds

3735

1613

96.40 %

32.7 min

150

3 seconds

4990

2149

98.09 %

35.56 min

100

2 seconds

7498

3223

95.03 %

32.4 min

50

1 second

15022

6448

93.24 %

32.6 min

20

0.4 second

18797

8063

80.14 %

33.1 min

12

0.24 second

31322

13440

68.58 %

33.3 min

Table 5. Accuracy testing for different data packet sizes

Table 5. Accuracy testing for different data packet sizes

Comment:

The section also talks about "making an estimate of  the value of gravity", which confused me, hence my comment on figure 2 querying why it isn't 9.8m^-2? The author should clarify that they are measuring the vertical component of instantaneous acceleration.

Answer:             

The whole section has been modified to clarify that we are estimating the direction of the force of gravity (not the value of the gravity acceleration but just an estimate of where its direction is in the sensor coordinate system).

For example in:

This procedure consists of making an estimate of the direction of the gravity force in a window with a duration of "2m" samples. It means that the gravity vector will be estimated at each instant, using the previous "m" samples and the "m" following samples at the instant in which the estimate is to be calculated.

Comment:

Figure 2 could be improved by labelling the axis with time (in seconds). The timestamp is less informative.

Answer:

The figure has been regenerated with the labelling in seconds:

Comment:

Page 10 - what are the "different test groups"

Answer:

We have provided the following text in order to better describe what and how the different groups were generated:

To better measure the performance of these ID1 and ID5 configurations, we took different test groups and calculated the average accuracy in different executions, obtaining the results in Table 2. A different seed was used in the random algorithm to split the data into 70% training and 30% validation, The seed values used are captured as identifiers in the first column in Table 2.

Comment:

Table 2: you could compute the mean and st dev of the measurements and assess how significant is the difference between the two columns?

Answer:

The value for the variance has been added in order to assess if a 1% improvement in accuracy is significant. The following text has been added:

The results in Table 2 show the mean value and the variance for each configuration and group. The variance for the different groups for configuration ID5 tends to be smaller while the mean value shows an improvement around 1% in accuracy.

Seed group test

Accuracy for Configuration ID 1

LSTM (50 - 100) & 90 - 100

Accuracy for Configuration ID 5

LSTM (100 - 100) & 90 -100

10

Mean: 97.82 %

Variance: 0.861

Mean: 98.35 %

Variance: 0.299

20

Mean: 96.23 %

Variance: 0.98

Mean: 97.61 %

Variance: 0.667

40

Mean: 97.08 %

Variance: 0.612

Mean: 98.15 %

Variance: 0.425

50

Mean: 96.76 %

Variance: 0.774

Mean: 96.97 %

Variance: 0.746

Table 2: Comparison of ID1 and ID5 architecture in different test sets

Comment:

Page 16: Is training time really important, after all, you only train once.

Answer:

We included the training time to give an idea about the complexity of the model in terms of how much time is required to train it. We have added the following information to the manuscript:

The results are shown in Table 5 (the training time has been included in the table to give an idea about the complexity of the model)

Comments in the pdf document:

We have included the proposed changes into the revised manuscript.

Reviewer 2 Report

The paper presents many results from a method based on a Deep recurrent neural network to identify the gait.

It is well written generally speaking. The introduction is clear and the state of the art complete. Results of the method from an open access database are presented and also expanded with others collected experimentally from two other volunteers.

I think the paper is valuable enough to be published but some improvements should be performed.

My main concern is the comparison with other methods. This certainly has to be improved and completed. It is briefly covered in the Conclusions section to say that the method outperforms others but very (very) slightly. Actually, in the Table 5 of your manuscript there is a difference of 1.69% in accuracy between a packet size of 150 and other of 200 and differences in performance larger than 1% are common in your results and seem to be justified by statistics even. An improvement of 0.1% (of averaged quantities) has to be justified as statistically significant and it is still small. Therefore, I suggest you defend your approach not only from these results but based on other aspects. In the same way, statements such as “ID5 configuration is more accurate that the ID1…” based on small differences (again averages and without evidence of statistical significance) should be moderated.

Other comments:

  • Section 3.1.1.: be careful with the name you give to the vector that results from averaging the accelerometer output. It can be confused with the gravity. I understand that they should be the same provided that the other linear accelerations are modeled as a noise with zero mean. Then the result of averaging is just the gravity vector. I think the explanation of this part should be improved a little.
  • Last three lines in page 8: I think the details of the memory units per layer could be better given later, once the results are obtained, but it is only an opinion. The way it is indicated in Figure 5 is not clear either and also the concept of “memory unit”. If you have 100 memory units means that you store data from the last 100 samples? Please explain this a little.
  • Please introduce the meaning of the parameters in page 9 (loss function, optimizer, metric)
  • Please explain the meaning of the first column in Table 2 (seed group test)
  • Please introduce earlier the meaning of “loss” in Figure 6 and 7
  • Confusion matrixes: I wonder why different values are obtained when the results for different true labels are summed up and compared.
  • Please improve the explanation of (4). Actually you only make the scalar product of the gyro output with the previously obtained g vector from acceleration data.

Other minor comments:

  • A native English speaker should review the English, though it is good generally speaking.
  • Please be careful with labels in the figures, many of them are in Spanish (i.e. “Aceleración” in vertical axis of Figure 2, “Entrenamiento” and “validacion” in Figures 6 and 7)
  • Please clarify the meaning of “instances of time” in page 7 (instants?)
  • Please clarify the meaning of the symbol “&” in Table 1, it would be even good if the explanation is in the caption.
  • Table 1: the word “parameters” is broken incorrectly
  • Page 12: last paragraph should be improved. For instance, I guess “capture capacity” is actually output data rate.

Author Response

Answer to the reviewers’ comments for the paper: Gait Based Identification Using Deep Recurrent Neural Networks and Acceleration Patterns

We would like to thank the reviewers for their positive feedback which has allowed us to improve the quality of the manuscript. All the proposed changes have been fully revised and thoroughly implemented.

Reviewer 2:

Comment:

My main concern is the comparison with other methods. This certainly has to be improved and completed. It is briefly covered in the Conclusions section to say that the method outperforms others but very (very) slightly. Actually, in the Table 5 of your manuscript there is a difference of 1.69% in accuracy between a packet size of 150 and other of 200 and differences in performance larger than 1% are common in your results and seem to be justified by statistics even. An improvement of 0.1% (of averaged quantities) has to be justified as statistically significant and it is still small. Therefore, I suggest you defend your approach not only from these results but based on other aspects. In the same way, statements such as “ID5 configuration is more accurate that the ID1…” based on small differences (again averages and without evidence of statistical significance) should be moderated.

Answer:

Apart from the accuracy values obtained, it is important to notice that the data capture has been carried out without the need of a controlled setting, allowing the users to freely walk in different conditions. We have added more information to better show our results in the context of previous research studies and to highlight the difference in the setting as:

The results presented in this manuscript show that deep Recurrent Networks are able to automatically characterize the gait properties and particularities for users when walking in the wild (without the need of a predefined setting) with accuracies higher than 0.97. The results outperform previous studies for gait characterization in unconstrained conditions [21] in which the accuracy values are bellow 0.94 using a hybrid deep neural architecture and previous RNN based architectures for gait inspection [22] in which the authors achieve accuracy values below 0.93. Although the accuracy of the proposed method in this manuscript only slightly outperforms other previous studies in controlled settings, it allows the walking activity to be freely executed by each user. The authors in [19] reported a 95% accuracy when the data is collected from 6 predefined walking segments. The authors in [20] obtained similar accuracy values from the ones presented in this manuscript but also using the same dataset in the same controlled environment for data capturing.

Comment:

  • Section 3.1.1.: be careful with the name you give to the vector that results from averaging the accelerometer output. It can be confused with the gravity. I understand that they should be the same provided that the other linear accelerations are modeled as a noise with zero mean. Then the resuClt of averaging is just the gravity vector. I think the explanation of this part should be improved a little.

Answer:

The whole section has been modified to clarify that we are estimating the direction of the force of gravity (not the value of the gravity acceleration but just an estimate of where its direction is in the sensor coordinate system).

For example in:

This procedure consists of making an estimate of the direction of the gravity force in a window with a duration of "2m" samples. It means that the gravity vector will be estimated at each instant, using the previous "m" samples and the "m" following samples at the instant in which the estimate is to be calculated.

Comment:

  • Last three lines in page 8: I think the details of the memory units per layer could be better given later, once the results are obtained, but it is only an opinion. The way it is indicated in Figure 5 is not clear either and also the concept of “memory unit”. If you have 100 memory units means that you store data from the last 100 samples? Please explain this a little.

Answer:

We have added “(as captured in section 4)” to better point to the section which provides the details.

We have also added the following text in order to better clarify the objective of the memory units inside LSTM cells:

Memory units represent the internal state of the LSTM cell and are able to learn and forget time dependent patterns based on the previous state and current inputs. The optimal number of memory units will therefore depend on the complexity of the time dependencies in the patterns to be detected.

Comment:

  • Please introduce the meaning of the parameters in page 9 (loss function, optimizer, metric)

Answer:

The parameters have been introduced as proposed by the reviewer:

  • Loss function (the function used to estimate how far the current parameters are from the optimal ones when training the model and therefore the function to be minimized in such training): Categorical cross entropy. Used for multi-class classification problems.
  • Optimizer (the numerical algorithm used to find the optimal value for the parameters in the model that minimize the loss function) : Adam. It has the advantages of other optimizers such as RMSprop and SDG (Stochastic Gradient Descent) with momentum. It offers fast convergence and its operation is based on the use of first and second moment gradient estimates, adapting its learning rate to each weight of the neural network.
  • Metric (the mechanism used to measure the achieved performance): Categorical accuracy. Accuracy of sample identification in multi-class classification problems.

Comment:

  • Please explain the meaning of the first column in Table 2 (seed group test)

Answer:

We have added an explanation in the text as:

A different seed was used in the random algorithm to split the data into 70% training and 30% validation, The seed values used are captured as identifiers in the first column in Table 2.

Comment:

  • Please introduce earlier the meaning of “loss” in Figure 6 and 7

Answer:

The meaning is added as:

Loss is the scalar value that tends to be minimized, since a lower value implies that the values predicted by the model are correct and correspond to the real labels of the samples. In this case, the categorical cross entropy loss function ("Categorical cross entropy") has been used to calculate the value of loss.

Comment:

  • Confusion matrixes: I wonder why different values are obtained when the results for different true labels are summed up and compared.

Answer:

We have captured the confusion matrixes in order to see how the model estimates similarities for each user with the rest of the users. If the samples which are wrongly classified for a particular user randomly correspond to other users, using several identification windows will just find the right user while if the misclassification for a user is always with the same user it will have different implications.

Comment:

  • Please improve the explanation of (4). Actually you only make the scalar product of the gyro output with the previously obtained g vector from acceleration data.

Answer:

We have modified the text as the reviewer points out:

The process is similar to that for acceleration data. First the direction of the gravity force is estimated by averaging the acceleration data as previously described. Then, the vertical rotation speed (the rotation on the direction of the estimated unit gravity force) is calculated using the dot product as:

(4)

Comment:

  • A native English speaker should review the English, though it is good generally speaking.

Answer:

We did not have the time to send the paper to a native English reviewer but we have done a review ourselves to try to improve the quality of the writing.

Comment:

  • Please be careful with labels in the figures, many of them are in Spanish (i.e. “Aceleración” in vertical axis of Figure 2, “Entrenamiento” and “validacion” in Figures 6 and 7)

Answer:

We have changed them:

Comment:

  • Please clarify the meaning of “instances of time” in page 7 (instants?)

Answer:

We have changed it for instants of time,

Comment:

  • Please clarify the meaning of the symbol “&” in Table 1, it would be even good if the explanation is in the caption.

Answer:

We have replaced it by “and” (the number of memory units in layer 1 and layer 2).

Comment:

  • Table 1: the word “parameters” is broken incorrectly

Answer:

done

Comment:

  • Page 12: last paragraph should be improved. For instance, I guess “capture capacity” is actually output data rate.

Answer:

We have modified it.
